# Emerging Roles and Mechanisms of RNA Modifications in Neurodegenerative Diseases and Glioma

**DOI:** 10.3390/cells13050457

**Published:** 2024-03-05

**Authors:** Ami Kobayashi, Yosuke Kitagawa, Ali Nasser, Hiroaki Wakimoto, Keisuke Yamada, Shota Tanaka

**Affiliations:** 1Department of Neurology, Brigham and Women’s Hospital, Harvard Medical School, Boston, MA 02115, USA; 2Department of Neurosurgery, Massachusetts General Hospital, Harvard Medical School, Boston, MA 02114, USA; ykitagawa@mgh.harvard.edu (Y.K.); amnasser@mgh.harvard.edu (A.N.); hwakimoto@mgh.harvard.edu (H.W.); 3Translational Neuro-Oncology Laboratory, Massachusetts General Hospital, Harvard Medical School, Boston, MA 02114, USA; 4Department of Neurosurgery, The University of Tokyo, Tokyo 113-0075, Japan; far.east.movement0913@gmail.com (K.Y.); stanaka@m.u-tokyo.ac.jp (S.T.); 5Department of Neurosurgery, Okayama University Graduate School of Medicine, Dentistry and Pharmaceutical Sciences, Okayama 700-8558, Japan

**Keywords:** RNA modifications, neurodegeneration, glioma

## Abstract

Despite a long history of research, neurodegenerative diseases and malignant brain tumor gliomas are both considered incurable, facing challenges in the development of treatments. Recent evidence suggests that RNA modifications, previously considered as static components of intracellular RNAs, are in fact dynamically regulated across various RNA species in cells and play a critical role in major biological processes in the nervous system. Innovations in next-generation sequencing have enabled the accurate detection of modifications on bases and sugars within various RNA molecules. These RNA modifications influence the stability and transportation of RNA, and crucially affect its translation. This review delves into existing knowledge on RNA modifications to offer a comprehensive inventory of these modifications across different RNA species. The detailed regulatory functions and roles of RNA modifications within the nervous system are discussed with a focus on neurodegenerative diseases and gliomas. This article presents a comprehensive overview of the fundamental mechanisms and emerging roles of RNA modifications in these diseases, which can facilitate the creation of innovative diagnostics and therapeutics for these conditions.

## 1. Introduction

Epigenetic modifications play crucial roles in the development, function, and plasticity of the nervous system. These modifications involve DNA methylation, histone modifications, chromatin remodeling, and RNA modifications. Epigenetic dysregulation has been linked to neurodegenerative diseases, brain tumors, and other various neurological disorders. Understanding the roles of epigenetic modifications in the nervous system is essential for gaining insights into normal brain function, neurological disorders, and potential therapeutic interventions. 

Among all epigenetic modifications, RNA modifications are generally considered distinct from classical modifications involving DNA and histones [1]. Recently, more studies have been reported on “epitranscriptomics”, which refers to a molecular biology field that focuses on the biochemical modifications of RNA molecules and their effect on gene expression and various cellular functions [2]. Epitranscriptomics research aims to understand the distribution patterns, biogenesis mechanisms, and regulatory functions of modified RNA molecules, as well as the interactome, evolutional conservation, and novel reader proteins. It has become evident that the information carried by RNA molecules is not static and can be dynamically modified by adding, removing, or modifying chemical groups [3]. The link between atypical RNA modifications and a range of neurological disorders highlights the significance of these chemical modifications in understanding the underlying mechanisms of these diseases. Regulating RNA modifications can provide new opportunities for therapies, as they offer a new area of biology to explore. In this review, we will outline the RNA modifications, including non-coding RNAs, and uncover their distinct functions and the consequential effects on cellular and physiological processes in the pathogenesis of neurologic disorders (Figure 1).

## 2. Modifications on mRNAs, rRNAs, and Non-Coding RNAs

Although numerous RNA modifications have been identified including messenger RNAs (mRNAs), ribosomal RNAs (rRNAs), and non-coding RNAs, only a handful of them have undergone thorough investigation within the nervous system. Some of the well-established RNA modifications are N6-methyladenosine (the methylation of adenosine at position 6, m6A), N1-methyladenosine (m1A), 5-methylcytosine (m5C), pseudouridine, and RNA editing. These modifications have been linked to various processes in the brain, including neurodevelopment, neurogenesis, neuroplasticity, learning and memory, neural regeneration, neurodegeneration, and brain tumorigenesis.

### 2.1. N6-Methyladenosine (m6A)

Since the 1970s when m6A modification was first identified [4], m6A became the most extensively researched and prevalent RNA modification in human cells [5]. With the advancement of identifying and detecting transcriptome-wide m6A distribution at single-base resolution [6,7], we now know that m6A modifications are crucial in various biological processes, including gene expression regulation, RNA processing, and RNA stability. These modifications are installed by methyltransferase complexes and can be reversed by demethylases, allowing them to have a dynamic and reversible nature. m6A marks have been discovered across different RNA repertoires: mRNAs, rRNAs, transfer RNAs (tRNAs), circular RNAs (circRNAs), and micro RNAs (miRNAs). Specifically for mRNAs, most m6A marks are situated at the start of the final exons, in the 3′-untranslated region (UTR), or near to the stop codons [8,9]. For long non-coding RNAs (lncRNAs), high-throughput methods revealed that human m6A marks were mapped to certain well-characterized lncRNAs, including X-inactive specific transcript (XIST) and metastasis-associated lung adenocarcinoma transcript 1 (MALAT-1) [10,11,12]. Writer complexes like Methyltransferase-like 3 and 14 (METTL3 and METTL14), Wilms tumor 1-associated protein (WTAP), Zinc finger CCCH domain-containing protein 13 (ZC3H13), RNA-binding motif protein 15/15B (RBM15/15B), and Vir-like m6A methyltransferase-associated protein (VIRMA) mediate the methylation of adenosine residues at position 6 [13,14,15,16,17]. METTL3 and METTL14 are heterodimers that form the core complex responsible for methyltransferase activity, stabilized by the association of another complex formed by WTAP, RBM15, ZC3H13, and VIRMA called the methylation-associated complex. METTL16 which is also one of the components of the m6A writer complexes, along with WTAP, is known to engage with specific lncRNAs, such as MALAT-1 [10]. YTHDF1, YTHDF2, YTHDF3, YTHDC1, YTHDC2, IGF2BP1, IGF2BP2, and IGF2BP3 have been identified as reader proteins that detect the m6A methylation marks. For instance, YTHDC1 facilitates the nuclear export of methylated mRNAs to the cytoplasm by specifically binding with m6A. Suppressing YTHDC1 leads to the nuclear buildup of m6A-modified mRNA, causing these transcripts to accumulate within the nucleus and become scarce in the cytoplasm [18]. In addition, it has been reported that m6A modifications in circRNAs promote efficient protein translation in a cap-independent fashion through YTHDF3 and the translation initiation factors eIF4G2 and eIF3A [19]. This m6A-driven translation by circRNAs can be enhanced by METTL3 and METTL14. Meanwhile, enzymes such as FTO and ALKBH5 act as erasers for removing m6A marks. m6A modifications can either increase or decrease mRNA stability, depending on the specific reader proteins that bind them. Functionally, m6A participates in nearly every stage of the mRNA life cycle, affecting splicing, other nuclear processing, and degradation within the cytoplasm [20].

m6A modifications are known to impact a range of brain functions and development, including neurogenesis, spinogenesis, learning and memory, dendritic structure, axon regeneration, and brain development. Some of the neurologic disease phenotypes are a consequence of disrupted m6A pathways, attributed to either disease-specific mutations or alterations in the levels of various m6A modulators. The m6A marks located in the 3′-UTR regions of mRNAs are believed to play significant mechanistic roles in the onset of age-related disorders such as Alzheimer’s disease (AD), Parkinson’s disease (PD), and Amyotrophic lateral sclerosis (ALS), as they regulate the translation of associated transcripts. In Table 1, we provide an overview of various RNA modifications, including m6A marks, alongside detection methods, and their connections to diverse RNA metabolic activities associated with neurological diseases. 

### 2.2. N1-Methyladenosine (m1A)

Methylation occurring at the N1 position of adenosine represents another example of dynamic modifications of RNA within mammalian systems [21]. This modification is abundant, prevalent, and conserved across both prokaryotic and eukaryotic RNAs as an internal post-transcriptional alteration, particularly prominent in the RNAs of higher eukaryotic cells [22,23]. This modification halts translation elongation, which is the process of translating the genetic code into a polypeptide chain. It also hinders the incorrect incorporation of nucleotides during reverse transcription (RT) by impeding the formation of Watson–Crick base pairs [24]. While m1A methylations have been reported in the 5′-UTR regions and coding sequences of mRNAs, they have been primarily identified in rRNAs and tRNAs, altering their structure and influencing their capacity for protein binding, stability, and functionality. The most abundant and representative m1A writer in the cytosol is the heterotetrameric tRNA methyltransferase TRMT6/61A, which incorporates a GUUCRA tRNA-like motif with a t-loop structure into specific mRNAs [25]. On the other hand, ALKBH3 and ALKBH1 serve as erasers for m1A, with ALKBH3 responsible for demethylating the m1A mark on mRNAs [26,27]. In addition, reports indicate that YTHDF2 not only serves as the reader for m6A but can also specifically detect m1A-modified sequences and bind to endogenous m1A-modified transcripts with a low affinity [28]. These findings suggest that YTHDF2 could play a more expansive role in RNA modifications beyond its established function as an m6A reader. Modifications of m1A in mRNA are primarily linked to the regulation of mRNA translation and decay, suggesting crucial roles in ensuring proper brain development and function. However, studies on the correlation between m1A marks and neurological disorders are still limited.

### 2.3. 5-Methylcytosine (m5C)

m5C modifications have been identified across a spectrum of RNA species, including rRNAs, tRNAs, and mRNAs, as well as small nuclear RNAs (snRNAs), vault RNAs, enhancer RNAs, lncRNAs, and miRNAs. These modifications are crucial in regulating RNA stability, transcription, transportation, and translation. They have been associated with gene expression and metabolism. For example, early studies have demonstrated that m5C modifications affect the interactions of lncRNAs, such as MALAT1, antisense non-coding RNA in the INK4 locus (ANRIL), nuclear paraspeckle assembly transcript 1 (NEAT1), growth arrest-specific transcript 5 (GAS5), ribonuclease P RNA component H1 (RPPH1), Pvt1 oncogene non-protein coding (PVT1), SNHG12, Telomerase RNA component (TERC), and XIST with chromatin-associated protein complexes [29,30]. In human cells, the NOL1/NOP2/SUN domain-containing protein family, specifically NOP2/SUN RNA methyltransferase (NSUN) 1 through NSUN7, along with the DNA methyltransferase (DNMT) homolog DNMT2, are responsible for generating m5C marks on RNA molecules [31]. NSUN1, NSUN2, and NSUN5 especially exhibit conservation across eukaryotes, while the remaining family members are exclusive to higher eukaryotes. In terms of functionality, NSUN1 and NSUN5 are involved in methylating cytosine C5 of rRNAs in the cytosol, while NSUN2, NSUN6, and DNMT2 are responsible for methylating cytosolic tRNAs. Additionally, NSUN2, NSUN3, and NSUN4 perform methylation at cytosine C5 within mitochondrial RNAs [32,33]. The mRNA export adaptor protein ALYREF and Y-box binding protein1 (YBX1) bind and recognize the m5C modifications and regulate the nucleo-cytoplasmic transport and stability of mRNA molecules [34,35]. The modifications of RNAs through m5C can be reversed by α-ketoglutarate (α-KG)-dependent dioxygenases, like ten-eleven translocation (TET)1 and TET2, which actively demethylate these marks [36].

A major significant role of m5C RNA methylation is that m5C RNA modifications affect the stability of eukaryotic rRNA by impacting the folding of crucial ribosomal regions, consequently controlling translation [37]. In addition, the modification at the m5C position can influence the aminoacylation step in translation, thereby impacting the overall accuracy of the translation process [38,39]. Notably, the impact of m5C modification on mRNA translation differs based on m5C placement. It hampers translation efficiency if it occurs within the 5′-UTR or coding sequence (CDS). However, m5C modification in the 3′-UTR, when mediated by NSUN2, has been found to increase translation efficiency [40,41,42,43]. The essential functional roles fulfilled by m5C modifications suggest that the disrupted expression of these genes could contribute to diverse neuronal conditions. For instance, a study that involved conditionally knocking out NSUN2 in the prefrontal cortex of mice resulted in bidirectional behavioral changes associated with depression [44]. The deficiency of NSUN2 led to modifications in almost 1500 proteins within the prefrontal cortex, accompanied by reduced translation efficiency linked to a glycine-codon defect. This, in turn, caused a disruption in synaptic communication among pyramidal neurons in the prefrontal cortex. Another example is the interruption of NSUN2’s tRNA methylation activity which can cause an accumulation of tRNA fragments at the 5′ ends, interfering with the formation of upper layer neurons and negatively impacting brain development in mice [45].

### 2.4. Pseudouridine

Pseudouridine, the isomer of uridine known as 5-ribosyl uracil or Ψ, is less abundant in mRNA; instead, it is more frequently detected in non-coding RNAs such as rRNAs, snRNAs, and tRNAs. Regardless of the abundance among mRNAs, pseudouridine can still influence the secondary structure of these molecules. For example, pseudouridine residues were found in MALAT1 at the positions of U5160, U5590, and U3374 [46,47]. However, the functions of these chemical alterations and their influence on MALAT1 activity remain unclear, requiring more comprehensive research to understand them. Two primary mechanisms for RNA pseudouridylation have been identified. First is the pseudouridylation that is independent of guide RNA and relies on pseudouridine synthase (PUS) enzymes directly catalyzing the transformation of uridine into Ψ within their target sequences (Table 1). On the other hand, pseudouridylation that is dependent on guide RNA involves H/ACA-box small nucleolar RNAs (snoRNAs) binding to target RNAs through sequence-specific interactions [48,49,50]. Codons containing Ψ have been demonstrated to exert a modest influence on ribosomes by incorporating specific amino acids. Additionally, stop codons containing pseudouridine (Ψ) have been found to guide the suppression of translation termination [51,52,53]. Furthermore, the existence of Ψ in numerous RNAs alters their interactions with RNA-binding proteins (RBPs) that participate in nuclear RNA processing, as well as the localization or stability of cytosolic RNA [48,49]. 

### 2.5. Adenosine to Inosine RNA Editing (A-to-I Editing)

RNA editing, a post-transcriptional modification, involves converting specific nucleotides into RNA molecules. This includes adenosine-to-inosine (A-to-I) editing, prevalent in vertebrates, with numerous such sites discovered in mice and humans. This editing process is orchestrated by the adenosine deaminase acting on RNA (ADAR) protein family. The adenosine deaminase that acts on tRNA (ADAT) conducts this conversion for tRNAs. The change from adenosine to inosine is due to the deamination of the amino group at adenosine’s C6 position, altering the RNA’s informational content and potentially its secondary structure [54,55]. In addition, A-to-I RNA editing is known to contribute significantly to the diversity of the epitranscriptome and the proteome in various cancers [56,57]. RNA editing plays a crucial role in modulating gene expression, particularly by altering miRNA expression, maturation, and stability [58,59,60]. This regulatory mechanism involves the activity of specific enzymes from the adenosine/cytidine deaminase family, which trigger single nucleotide transformation in primary miRNAs. Such modifications can substantially affect the miRNA’s stability and its ability to mature and target specific mRNAs, ultimately influencing miRNA-guided gene expression regulation. Additionally, RNA editing modifies the primary structure of wild-type mature miRNAs, thereby altering their gene-regulatory functions.

RNA editing, particularly A-to-I editing, is an extensive process in humans, with estimates suggesting that up to 85% of pre-mRNAs are subject to this modification [61,62,63]. This editing process predominantly occurs within introns and untranslated regions (UTRs) of genes encoding proteins and is reportedly involved with neural development and various neurological disorders. For example, RNA editing at the I/V site of the potassium channel Kv1.1, mediated by ADAR2, has been associated with epilepsy [64]. Episodic ataxia type 1, a condition characterized by seizures, ataxia, and myokymia, has been reported to be caused by mutations in the gene KCNA1, which codes for Kv1.1. In addition, ADAR2’s role in editing the RNA of glutamate receptors is also associated with psychiatric diseases, including schizophrenia [65] (Table 1). 

### 2.6. Other Modifications

An extensive array of RNA modifications has been discovered, particularly in tRNAs, which includes m5C, 3-methylcytidine (m3C), 1-methyl-guanosine (m1G), N2-methylguanosine (m2G), N7-methylguanosine (m7G), N2,N2-dimethylguanosine (m22G), N1-methylpseudouridine (m1J), 2-methyladenosine (2 mA), 5-formyl-20-O-methylcytidine (f5Cm), 3-(3-amino-3-carboxypropyl) uridine (acp3U), 5-methyluridine (5 mU), 5-methoxycarbonylmethyluridine (mcm5U), 5-methoxycarbonylmethyl-2-thiouridine (mcm5s2U), dihydrouridine, queuosine (Q), galactosylqueuosine (gal Q), mannosyl-queuosine (manQ), N6-threonylcarbamoyladenosine (t6A), N6-methyl-N6-threonylcarbamoyladenosine (m6t6A), 2-methylthio-N6-threonylcarbamoyladenosine (ms2t6A), N4-acetylcytidine (ac4C), N6-isopentenyladenosine(i6A), peroxywybutosine (o2yW), and wybutosine (yW) [66,67]. These modifications are categorized by their position into two principal types: some that are located in the anticodon loop and others found elsewhere. The modifications occurring within the anticodon loop can influence translation by altering the base pairing between the codons in mRNAs and the tRNA that carries amino acids [68]. Modifications situated outside of the anticodon loop primarily influence the secondary structure of tRNAs.

Modifications in rRNA are relatively sparse, with just 2% of nucleotides undergoing changes. Among these, 2′-O-methylation (2′-OMe), where a methyl group is added to the 2′ hydroxyl of ribose, is the most common [69]. These 2′-OMe modifications can occur at the first nucleotide transcribed (denoted as m7GpppNmN-) or the second (m7GpppNmNm-). It has been shown that 2′-OMe modifications enhance the stability of RNA–RNA hybrid duplexes [70].

### 2.7. Detection and Quantification of RNA Modifications

To elucidate the dynamics of a broad spectrum of RNA modifications, including m6A, m1A, m5C, A-to-I editing, and modifications in tRNA, various sophisticated techniques are employed. Liquid chromatography–tandem mass spectrometry (LC-MS/MS) is pivotal for identifying and quantifying these modifications by leveraging the distinct mass changes in modified nucleotides [8,71]. Methylated RNA immunoprecipitation sequencing (MeRIP-Seq) specifically maps the genome-wide distribution of m6A, capturing m6A-modified RNA fragments [9]. RNA bisulfite sequencing (RNA-BisSeq) differentiates m5C from unmodified cytosine [29], while Ψ-seq specifically labels pseudouridine for precise localization [47]. Additionally, A-to-I RNA editing, mediated by ADAR enzymes, is identified through sequencing technologies that detect inosine as guanosine, due to its base-pairing properties [72]. The complexity of tRNA modifications, critical for the stability, structure, and function of tRNA, is revealed through methods such as LC-MS/MS and specialized sequencing techniques [73].

The regulatory roles of m6A methylation enzymes, including writers, readers, and erasers, are clarified by unveiling the regulation of gene expressions through quantitative Real-Time PCR (qRT-PCR) measuring mRNA levels. This is further investigated through immunoprecipitation coupled with mass spectrometry (IP-MS) and gene editing using CRISPR/Cas9 for gene knockout or knockdown, providing insights into the composition and interactions of these protein complexes. IP-MS elucidates the composition and interactions of these protein complexes [14]. Moreover, next-generation sequencing (NGS)-based detection techniques, such as MeRIP-Seq and Cross-linking and immunoprecipitation sequencing (CLIP-seq), alongside nanopore direct RNA sequencing, offer comprehensive methods for analyzing the regulation of RNA by m6A modifications, A-to-I editing patterns, and tRNA modifications, enriching our understanding of RNA biology in disease [71].

## 3. RNA Modifications in Neurodegenerative Diseases

### 3.1. RNA Modifications in Alzheimer’s Disease (AD)

One of the most abundant RNA modifications in the brain is m6A, and most of the recent studies on RNA modifications indicate that the loss of m6A methylation of RNA could promote AD development. Utilizing m6A-sequencing in combination with high-throughput LC–MS/MS, Shafik et al. revealed a substantial reduction in the expression level of METTL3, along with decreased m6A levels, in 5xFAD mice compared to control groups [74]. This result was consistent with other groups’ studies where notable reductions in both neuronal m6A methylation and METTL3 expression have been observed in human AD brains compared to those without the condition, as demonstrated by immunoblot analysis [75]. For circRNA, high-throughput sequencing has revealed significant changes in circRNA m6A methylation in APP/PS1 AD mice compared to control groups [76]. This result was consistent with another group’s study where METTL3-dependent m6A-modified circular RNA, circRIMS2, was significantly upregulated in APP/PS1 AD mice, which mediated synaptic and memory impairments by activating the ubiquitination of the GluN2B subunit of the NMDA receptor [77]. Silencing METTL3 or hindering the GluN2B ubiquitination by a short membrane-permeable peptide significantly rescued synaptic impairment in APP/PS1 AD mice. On the other hand, METTL3 knockdown in the mouse hippocampus has been associated with several negative outcomes, including memory loss, neurodegeneration, spine loss, and gliosis [75]. In terms of mechanism, the lack of METTL3 slows down the mRNA degradation of m6A-modified cell cycle genes like Cyclin D1 and Cyclin D2. This effect was observed in cultivated primary neurons, which resulted in impaired cell cycle control [75]. Furthermore, a recent investigation into the gene expression patterns regulated by m6A in post-mortem brains from AD patients revealed the abnormal expression of METTL3 and the RNA binding motif protein 15B in the hippocampus of individuals with AD. The study indicated that the accumulation of METTL3 in insoluble fractions exhibited a positive correlation with Tau levels in hippocampal lysates. This suggests that disruptions in m6A signaling might contribute to neuronal dysfunction in AD [78]. It is also found that the demethylase FTO triggers mTOR signaling and eventually activates Tau phosphorylation in AD [79]. Notably, the neuron-specific knockout of the gene FTO has been demonstrated to alleviate cognitive impairments in 3xTg AD mice [79]. Moreover, a genetic variant within the FTO locus was significantly linked to Alzheimer’s in the NIA-LOAD study [80,81]. The decreased expression of FTO in the cortex and amygdala of Alzheimer’s patients compared to healthy individuals indicates the functional significance of FTO in AD [80,81]. This is further substantiated by a prospective study indicating that individuals with the AA genotype in the FTO gene face an elevated risk of developing AD and other types of dementia [82]. 

Other reports demonstrate alterations in small RNA modifications in AD patients compared to healthy individuals. LC–MS/MS analysis showed an increase in m7G, 2′-O-methylcytidine (Cm), and 2′-O-methylguanosine, while m22G and N2,N2,7-trimethylguanosine (m2,2,7G) exhibited significant decreases in the miRNA fraction from the cortex of AD brains [83]. Interestingly, within small RNA fractions of rRNA-derived small RNAs, tRNA-derived small RNAs, Y RNA-derived small RNAs, and other unannotated RNAs, elevated levels of modifications such as Cm, 2′-O-methyluridine (Um), and m7G were observed in comparison to controls. Conversely, modifications such as pseudouridine, m1G, and m2,2,7G were diminished [83]. Additionally, microfluidic high-throughput PCR-based next-generation sequencing has discovered a significant reduction in A-to-I RNA editing levels in AD patients as compared to control samples [84]. In the pre-frontal cortex of individuals with AD, RNA editing of the GluA2 subunit of the AMPA receptor leads to alterations in intracellular Ca^2+^ levels, which are associated with neuronal dysfunction and neurodegeneration stemming from increased Ca^2+^ permeability [85]. In healthy individuals, less than 0.1% of all GluA2 RNA molecules are unedited in the pre-frontal cortex, contrasting with 1.0% in AD patients. Another study observed reduced RNA editing at the glutamine/arginine (Q/R) site of the Glu2 subunit in the caudate nucleus and hippocampus of sporadic AD patients who carry the Apo E4 allele [86]. The presence of the ApoE E4 allele, a known genetic risk factor for AD, is suggested to affect the AMPA receptor dynamics and glutamate regulation in the hippocampus [87,88,89]. Comprehensive analysis from the ROSMAP, MayoRNAseq, and MSBB studies identified millions of RNA editing sites across nine brain regions, with 108,010 edits to facilitate and 26,168 edits to impede the progression of AD [90]. 

### 3.2. RNA Modifications in Parkinson’s Disease (PD)

Individuals with PD display incapacitating motor impairments characterized by resting tremors, bradykinesia (reduced movement speed), muscle stiffness, and postural instability, commonly recognized as PD’s four cardinal manifestations. The emergence of these symptoms is attributed to the degeneration of dopaminergic neurons in the substantia nigra region of the brain. Some researchers have performed comprehensive genetic analyses, including genome-wide association, differential gene expression, and expression quantitative trait locus analyses, which have identified five m6A SNPs associated with altered gene expressions related to PD [91]. In one study, m6A methylation was reduced and FTO was highly expressed in PC12 cells, a cellular model of PD [92]. The overexpression of FTO in dopaminergic neurons decreases mRNA m6A modification and upregulates ionotropic glutamate receptor 1 (NMDAR1). This escalation contributes to oxidative stress, an increase in calcium influx, and ultimately, it accelerates the degeneration or cell death of these dopaminergic neurons [92]. In another study, METTL3, METTL14, and YTHDF2 were significantly downregulated in PD patient blood mononuclear cells with METTL14 being the main factor involved in the abnormal m6A modification of α-synuclein mRNA [93]. Mettl14 targets and regulates the expression of the α-synuclein gene by binding an m6A motif in the coding region which eventually modifies α-synuclein mRNA and weakens its stability. Spearman correlation analysis showed METTL14 levels inversely correlated with plasma α-synuclein concentrations and motor function of PD patients [93]. Furthermore, the advancement in RNA sequencing capabilities has facilitated the detailed exploration of RNA editing occurrences across the whole transcriptome, offering deeper insights into A-to-I editing in various diseases. A comprehensive transcriptome study linked PD to alterations in Alu insertions, which are the main substrates for the ADAR protein family [94]. A compelling and innovative concept has recently been suggested for leveraging the inherent editing activity of endogenous ADAR2 to correct a disease-causing mutation in PINK1 associated with PD [95]. This mutation, a G-to-A substitution, leads to a premature stop codon that truncates the PINK1 protein, resulting in the truncation of the protein’s C-terminus [95]. By designing guide RNAs that direct ADAR2 to target and edit the specific mRNA, researchers were able to restore PINK1/Parkin-mediated mitophagy in cell models [95].

### 3.3. RNA Modifications in Amyotrophic Lateral Sclerosis (ALS)

ALS, one of the prevalent neurodegenerative diseases, is marked by a progressive decline in muscle strength and atrophy, leading to both upper and lower motor neuron dysfunction. Many studies exploring the pathogenesis of ALS, where RNA modifications play a role, have demonstrated the involvement of A-to-I RNA editing. ADAR2, an RNA editing enzyme, is mislocated in cases of ALS/FTD caused by C9orf72 repeat expansion [96]. In mice with a conditional knockout of ADAR2, the Q/R site on GluA2, which is crucial for the proper functioning of the AMPA receptors in motor neurons, remains unedited, leading to gradual motor neuron degeneration [97,98]. Interestingly, ADAR2 is uniquely reduced in the motor neurons of ALS patients among all the ADAR family [99]. This led to abnormal Ca^2+^ influx through AMPA receptors in neurons, resulting in motor neuron degeneration, which is a characteristic feature of ALS [100,101,102]. Furthermore, TAR DNA-binding protein 43 (TDP-43), a nuclear RNA-binding protein implicated in both familial and sporadic forms of ALS, is specifically expressed in motor neurons that lack ADAR2. This suggests that unedited GluA2 at the Q/R locus is pathogenic in ALS [103]. Some reports suggest that the entry of Ca^2+^ through AMPA receptors that include unedited GluA2 results in calpain activation, which initiates TDP-43 pathology and deficits in nucleocytoplasmic transport, alongside causing excitotoxicity [104,105]. In other studies, the presence of Q/R site-unedited GluA2 followed by the downregulation of ADAR2 has been observed in ALS patients with mutation in fused in sarcoma (*FUS*^P525L^ mutation) with alterations of several circRNA expression levels [106,107]. 

## 4. RNA Modifications in Glioma

Gliomas account for nearly 80% of malignant brain tumors in adults [108]. They are broadly defined but generally are known to develop from the support cells of the brain called glia. The glial cells of the brain that most commonly undergo gliomagenesis are astrocytes, oligodendrocytes, and ependymal cells. Even with the standard of care, which includes a combination of surgery, radiation, and chemotherapy, gliomas recur at a high rate, which leads to a low overall survival (OS) for patients. The 2021 WHO classification implemented a new grading system for gliomas that integrates histopathological features and molecular diagnostic criteria to classify CNS tumors into more distinct subtypes further, improving approaches to patient care [109]. Although this improves diagnosis and allows for more targeted therapeutic approaches, patient outcomes remain poor, necessitating further molecular characterizations of these tumors to inspire novel treatments.

Epi-transcriptomics has been shown to be involved in the progression and malignancy of various cancer types, including gliomas [110]. m6A RNA methylation is the most heavily studied form of RNA modification. Various studies have shown that m6A RNA modification could inhibit or promote tumor progression depending on factors such as whether the target gene is an oncogene or a tumor suppressor and which components of the m6A methylation regulators are at play. In this section of the review, we explore the various ways that m6A methylation and its regulators, known as “Writers”, “Erasers”, and “Readers”, play roles in the emergence and progression of gliomas and the potential therapeutic approaches that target m6A methylation and its regulators.

### 4.1. m6A Methylation Regulators “Writers” in Glioma Pathogenesis and Treatment Resistance

METTL3 is a key component of the m6A methyltransferase complex, also comprising METTL14, and is crucial in the m6A methylation of nuclear RNA. It binds specifically to S-adenosyl methionine (SAM), facilitating m6A methylation, while METTL14 plays a supportive role despite its lack of a catalytic site [14,111]. Research on METTL3’s function in glioma has produced varied and sometimes conflicting results. Cui et al.’s initial studies showed that METTL3 knockdown results in increased growth, self-renewal, and tumorigenesis in glioma stem cells (GSCs), which are critical for the growth, invasion, and drug resistance in glioblastoma [112]. Conversely, other research indicates that silencing METTL3 or enhancing its mutated version suppresses GSC growth and proliferation. METTL3 downregulation reduced m6A methylation of serine- and arginine-rich splicing factors (SASFs), leading to YTHDC1-dependent nonsense-mediated decay of SASFs mRNA and lowered protein levels. Additionally, METTL3 influences the splicing of BCL-X and NCOR2, which play roles in cancer cell death and motility [113]. In contrast, Visvanathan et al. observed that overexpressing METTL3 correlates with enhanced GSC stemness and reduced differentiation, with the knockout of METTL3 or METTL14 increasing GSC sensitivity to gamma-irradiation [114]. These discrepancies may be due to different m6A methylation target genes and the specific “readers” recognizing these modifications.

METTL3 is involved in the methylation of non-coding RNAs, including lncRNAs and circRNAs, which are critical in glioma initiation and progression. One study demonstrated that METTL3-mediated methylation of LINC00839, regulated by YTHDF2, enhances its expression in GSC lines. This stabilization activates the Wnt/β-Catenin signaling pathway, increasing radioresistance and GSC proliferation [115]. Additionally, METTL3-induced m6A methylation of LINC01003 was found to regulate the focal adhesion kinase (FAK) pathway, promoting glioma cell proliferation and migration [116]. For circRNAs, Wu and colleagues found that METTL3′s m6A modification process increases the stability and expression of circDLC1 [117]. This elevation aids circDLC1 in its competitive binding with miR-671-5p, which in turn supports the transcription of Catenin Beta Interacting Protein 1 (CTNNBIP1) and ultimately inhibits the excessive growth of glioma cells.

METTL3-mediated m6A methylation is associated with treatment resistance in gliomas. Shi et al. reported that METTL3 methylation is elevated in glioblastoma tissues resistant to temozolomide (TMZ), with METTL3 overexpression increasing the stability of DNA repair enzymes, suggesting a mechanism for TMZ resistance [118]. Additionally, it has been shown that METTL3-mediated methylation indirectly fosters TMZ resistance by stabilizing Oxidized Low-Density Lipoprotein Receptor 1 (OLR1), a receptor involved in lipid metabolism and cellular signaling, and by activating the Wnt/β-Catenin pathway [119].

miR-1208 targets METTL3’s 3′UTR region, diminishing NUP214 levels, and inhibiting glioma cell proliferation [120]. Knockdown of the histone methyltransferase SETD2, which influences m6A modification, results in decreased levels of METTL3/14 and WTAP, reducing glioma cell proliferation and migration [121]. The resistance of glioma cells to mTOR inhibitors is attributed to m6A methylation at IRES sites, enhancing the translation of oncogenes, a process disrupted by METTL3/14 knockout [122]. Under fear stress, increased METTL3 expression stabilizes FSP1 through m6A methylation and inhibits ferroptosis in glioma cells, indicating a role in stress responses [123].

Another central component of the m6A methyltransferase complex is WTAP that has been implicated in various aspects of glioma progression and treatment response. One study found that Flotilin-1 (FLOT1), upregulated in gliomas and associated with advanced progression and poor prognosis, is stabilized by m6A methylation, with WTAP acting as the writer. The silencing of FLOT1 led to reduced glioma cell proliferation, highlighting the significance of WTAP in glioma biology [124]. Additionally, WTAP has been strongly linked to microsatellite instability, indicative of a compromised DNA mismatch repair system. Together with other genes like TRMT6, DNMT1, and DNMT3B, WTAP can predict overall survival in glioma patients and is correlated with poor post-operative outcomes [125]. Increased WTAP expression in glioblastoma compared to normal tissue further underscores its role in glioma pathogenesis [126].

WTAP also plays a role in the regulation of cell proliferation, migration, and apoptosis. A study demonstrated that miR-29a, which is typically underexpressed in GSCs, can inhibit the Quaking gene isoform 6 (QKI-6) and subsequently reduce WTAP expression. This inhibition leads to decreased activity in key pathways like phosphoinositide 3-kinase/AKT and extracellular signal-related kinase, reducing cell proliferation and invasion while promoting apoptosis [127]. Furthermore, WTAP expression levels have been closely correlated with tumor grading levels [128].

The modulation of WTAP expression can directly impact glioma tumorigenicity. Knockdown and overexpression studies have shown that WTAP can regulate epidermal growth factor receptor, influencing tumorigenicity in glioma. This was further confirmed in xenograft models [129].

Other m6A writers in glioma involve KIAA1429, RBM15, and ZC3H13. KIAA1429, also known as VIRMA, is consistently upregulated in glioblastoma and negatively correlated with the response to anti-cancer drugs, suggesting its role in drug resistance and tumor progression [130]. RBM15 is implicated in the proneural to mesenchymal transition (PMT) in GSCs. Studies have shown that neuronal activation can lead to changes in GSC behavior via miR-184-3p mediated inhibition of RBM15 expression, thus impacting radioresistance and progression. RBM15 knockdown led to decreased m6A modification and DLG3 mRNA levels, which in turn increased p-STAT3, a key signaling molecule in PMT [131]. Additionally, RBM15 expression has been shown to have prognostic value in glioma, particularly in predicting overall survival in patients with low-grade glioma (LGG) [132,133]. ZC3H13 plays a role in glioblastoma progression through its interaction with the tumor microenvironment. Under hypoxic conditions, ZC3H13 expression in microglia is influenced by neuron-derived exosomes, leading to changes in microglial polarization and subsequently affecting glioblastoma progression [134]. A prognostic model has indicated that ZC3H13 levels are positively associated with glioblastoma prognosis, suggesting its potential as a tumor suppressor [126]. Furthermore, the knockdown of ZC3H13 has been linked to increased TMZ resistance in Rb1 mutant glioblastoma cells [135]. In advancing glioma treatment, investigating m6A RNA methylation’s role in GSC biology, especially regarding stem cell maintenance and drug resistance, is a promising avenue. Such research would entail studying the impact of altering m6A writers such as METTL3, WTAP, VIRMA, RBM15, and ZC3H13 on GSCs. Employing CRISPR-Cas9 gene editing or RNA interference, modifications in m6A writers could be analyzed for their effects on GSC characteristics, including proliferation, differentiation, and survival under chemotherapy and radiotherapy. Although small molecule inhibitors of these writers have been designed and found to have promising results in other cancers, such as acute myeloid leukemia, they have not been heavily investigated in gliomas. These investigations are expected to uncover new molecular pathways affected by m6A modifications, leading to novel therapeutic targets that could effectively prevent glioma recurrence and contribute to improved glioma treatments.

### 4.2. m6A Methylation Regulators “Erasers” in Glioma Pathogenesis and Treatment Resistance

The role of erasers in m6A methylation, particularly in the context of glioma, is vital for understanding their influence on tumor progression and therapeutic resistance. Particularly, the erasers FTO and ALKBH5 exhibit a range of activities impacting various aspects of glioma biology. FTO, known for its role in fat consumption and overall metabolic rate regulation [136], also acts as a demethylase for m6A, affecting pre-mRNAs’ alternative splicing and 3′-end processing [137]. FTO’s influence extends to mitochondrial functions, affecting the expression of genes like SDHA and regulating the STAT3/FTO axis [138]. Prognostic studies suggest that FTO can predict poor outcomes in glioma patients, and inhibitors targeting FTO have shown promise in reducing tumorigenicity and aggressiveness in glioma models [139,140,141,142]. FTO expression is also linked to the decreased apoptosis and increased proliferation of glioma cells [143], and its inhibition suppresses GSC growth and self-renewal [112]. Intriguingly, FTO knockdown affects the nuclear localization of FOXO3a, influencing the expression of target genes like BIM, BNIP3, and BCL-6 [144].

FOXM1 is a transcription factor essential for tumorigenicity and invasion [145]. Silencing ALKBH5 suppresses FOXM1 and GSC proliferation, and its inhibitors have shown effectiveness in decreasing glioma cell proliferation [146,147]. ALKBH5 overexpression in glioblastoma stem cells contributes to increased resistance to radiation and enhanced invasive capabilities, and its activity is regulated by EGFR signaling, impacting ferroptosis through m6A modulation [148,149]. ALKBH5 also promotes PYCR2 expression, influencing glioma cell proliferation, migration, and PMT [150]. The regulation of ALKBH5 by USP36 underscores its role in glioblastoma progression and sensitivity to TMZ [151]. Its involvement in the proliferation, migration, and invasion of glioma cells has been well-established [152,153], and inhibitors targeting ALKBH5 reduce glioma cell migration and invasiveness [154]. Additionally, ALKBH5 influences immune responses in glioma, affecting cytokine expression and Programmed Death-Ligand 1 (PD-L1) protein levels [155]. The hypoxia-induced activity of ALKBH5 stabilizes transcripts like NEAT1, facilitating tumor-associated macrophage recruitment and immunosuppression, and enhances glioma cell growth by activating pathways like PPP [156,157]. In the context of TMZ resistance, ALKBH5 demethylates transcripts like NANOG, contributing to the development of resistance, and regulates TMZ sensitivity by interacting with transcripts like SOX2 [158,159].

Given the roles of FTO and ALKBH5 in modulating cell death in glioma, focusing on how these m6A erasers influence apoptosis and ferroptosis, especially in GSCs, could be valuable. Exploring the interplay between apoptosis inhibition in FTO and ferroptosis resistance in ALKBH5, particularly under radiotherapy, using gene editing techniques, offers promising insights. Such research could pave the way for innovative treatments aimed at overcoming radio-resistance and curbing GSC proliferation, ultimately improving the efficacy of glioma therapies. Moreover, further investigation is needed for inhibitors of these erasers to be fully established as promising treatment options.

### 4.3. m6A Methylation Regulators “Readers” in Glioma Pathogenesis and Treatment Resistance

In the intricate network of glioma pathogenesis, the YTHDF family, consisting of YTHDF1, YTHDF2, and YTHDF3, and YTHDC1 and YTHDC2, as members of the YTH domain-containing family, play significant roles in the prognosis and molecular mechanisms underlying glioma [158]. The YTHDC1 protein is more nuclear-focused, influencing mRNA splicing and export, whereas the YTHDC2 and YTHDF proteins are primarily involved in controlling the stability, degradation, and translation of m6A-modified mRNAs in the cytoplasm [160,161,162,163].

YTHDF1 is involved in various mechanisms that contribute to glioma resistance and progression. For instance, it stabilizes OLR1 mRNA, influencing the OLR1-mediated Wnt/β-Catenin pathway activation, which is linked to TMZ resistance in glioma [119]. Additionally, YTHDF1 plays a role in RNA editing, as it binds and promotes the translation of m6A-modified ADAR1, a molecule implicated in glioma progression [164]. YTHDF1’s expression, increased by C-myc overexpression, leads to higher levels of FDX1, a protein associated with several cancer signaling pathways [159]. Moreover, the protein Musashi-1, overexpressed in glioblastoma, upregulates YTHDF1, thereby sensitizing glioblastoma cells to TMZ and inhibiting their proliferation [165]. YTHDF2 is particularly significant in maintaining oncogene expression in glioblastoma stem cells. GSCs express YTHDF2 preferentially, and its targeting can inhibit cell growth and viability, suggesting the potential of the YTHDF2-MYC-IGFBP3 axis as a therapeutic target [156]. Furthermore, YTHDF2’s role extends to immune regulation, where its deficiency impairs the stability of ZDHHC3 mRNA, affecting PD-L1 expression and degradation in glioma [148]. YTHDF2 is also involved in key signaling pathways, including the receptor tyrosine kinase MET pathway, which is essential for glioblastoma stem cell renewal and tumorigenicity [166]. Its expression correlates with various immune cells in low-grade gliomas, and it has been shown to enhance TMZ resistance in glioblastoma [167,168]. Additionally, YTHDF2 promotes glioblastoma cell proliferation and tumorigenesis, largely through the downregulation of LXRα and HIVEP2 [169]. YTHDF3 has emerged as a potential target for treating Osimertinib-resistant glioblastoma cells. Research indicates that silencing YTHDF3 increases the sensitivity of these cells to Osimertinib, affecting their migratory and sphere-forming abilities [157]. Moreover, YTHDF3’s role in mTOR inhibitor resistance involves the methylation of IRES-mediated mRNA translation, highlighting its importance in resistance mechanisms [122].

On the other hand, YTHDC1 has been determined to be a prognostic marker for overall survival in patients with low-grade glioma (LGG) [133]. It plays a role in the regulation of key molecules and pathways that influence glioma cell behavior. For instance, YTHDC1 interacts with the circRNAs from the EPHB4 gene, which is implicated in glioma. METTL3-mediated m6A methylation of CircEPHB4 leads to its recognition by YTHDC1, which then localizes the transcript to the cytoplasm. This localization facilitates the stabilization of SOX2 mRNA, promoting the transcription of PHLDB2, a molecule associated with epithelial–mesenchymal transformation in various cancers. This mechanism underscores the role of YTHDC1 in enhancing stemness in glioma spheres, as evidenced by the increased expression of stemness marker proteins and the promotion of cell proliferation, invasion, and migration [170,171,172]. Additionally, YTHDC1 influences glioma cell proliferation through its impact on VPS25, a protein upregulated in gliomas. Knockdown of YTHDC1 leads to decreased VPS25 levels and reduced cell proliferation [173]. Furthermore, YTHDC1 is implicated in the non-sense mediated mRNA decay (NMD) of SASFs, a process dependent on the m6A methylation status of these factors [113]. YTHDC2, similarly, is crucial in the context of LGG. It is overexpressed in these tumors and correlates with patient prognosis, where higher levels of YTHDC2 are associated with poorer outcomes [174]. Moreover, YTHDC2 has been identified as an independent negative prognostic indicator for overall survival in gliomas. This finding, derived from Cox regression multivariate analysis, suggests that YTHDC2 levels could be a significant marker in evaluating the progression and potential outcomes of glioma treatments [175]. YTHDC1 and YTHDC2 are integral components in the molecular landscape of glioma, influencing various aspects of tumor biology. Their roles in RNA processing, interaction with circRNAs, and impact on key signaling pathways underscore their potential as prognostic biomarkers and as targets for therapeutic intervention in glioma.

HNRNPC and HNRNPA2B1 are extensively expressed m6A regulators in the tumor microenvironment, suggesting their importance in a variety of tumor-relevant cell types [176]. HNRNPC shows significant expression in glioblastoma, serving as an essential splicing factor. Higher expression levels in glioblastoma tissues compared to normal tissues are associated with a poorer prognosis in high-risk patients. Additionally, HNRNPC expression positively correlates with PD-L1, highlighting its prognostic significance and role in immune modulation [166]. HNRNPC interacts with long non-coding RNA DDX11 antisense RNA 1 (DDX11-AS1) to promote the Wnt/β-Catenin and AKT pathways, influencing the epithelial–mesenchymal transition (EMT) and glioma cell migration. Knockdown of HNRNPC disrupts these pathways and hinders EMT. Furthermore, HNRNPC regulates microRNA-21 (miR-21) expression, thereby affecting glioblastoma progression via Programmed Cell Death 4 (PDCD4) [177,178]. HNRNPA2B1 is identified as an independent prognostic factor for glioma. HNRNPA2B1 promotes GSC self-renewal and tumorigenesis by modulating cholesterol biosynthesis, notably through the stabilization of SREBP2 mRNA. This stabilization boosts the expression of HMGCR and LDLR mRNA, key components in cholesterol biosynthesis and reuptake. Combining HNRNPA2B1 suppression with cholesterol metabolism drugs yields potent inhibitory effects on glioma cells [179]. HNRNPA2B1 also plays a role where circular RNA from the NEIL3 gene (known as circNEIL3) is packed into exosomes and transferred to tumor-associated macrophages within the tumor microenvironment [180]. This transfer promotes the suppression of the immune response by stabilizing IGF2BP3, thereby facilitating glioma progression. HNRNPA2B1 modulates cell cycle dynamics, apoptosis, and treatment responses, particularly to β-asarone. Knockdown of HNRNPA2B1 reduces cell proliferation and increases apoptosis, affecting signaling pathways such as AKT and STAT3. This modulation alters the expression of proteins like B-cell lymphoma-2 (Bcl-2), CyclinD1, and PCNA, underscoring the role of HNRNPA2B1 in glioma cell proliferation and its therapeutic potential [181,182,183].

The Insulin-like Growth Factor 2 mRNA-Binding Proteins (IGF2BP) family, comprising IGF2BP1, IGF2BP2, and IGF2BP3, plays a crucial role in glioma development and response to treatment. IGF2BP1 is upregulated in mesenchymal glioblastoma compared to proneural glioblastoma, correlating with poor patient outcomes. Its overexpression in proneural glioblastoma increases cell stemness, while its knockdown in mesenchymal glioblastoma leads to decreased proliferation and sphere formation. IGF2BP1 specifically binds m6A on YAP mRNA to stabilize it, increasing protein expression and activating the YAP/TAZ complex involved in the Hippo signaling pathway. This indicates a feed-forward loop enhancing tumorigenicity and stemness, particularly in mesenchymal glioblastoma [184]. Furthermore, IGF2BP1 is targeted by various non-coding RNAs and microRNAs, such as LINC00689, PCAT6, Lnc-THOR, and miR-4500, which regulate its expression and thus impact glioma progression and apoptosis [185,186,187,188,189,190]. IGF2BP2’s involvement in glioma includes interactions with various RNA elements and transcription factors. For instance, it participates in vascular mimicry through its interaction with HOTAIRM1 [191] and engages in the LINC00265/miR-let-7d-5p/IFI30/ZNF384/IGF2BP2 axis to regulate EMT and stemness [192]. The protein complex formed between HOXD-AS2 and IGF2BP2 is associated with poorer prognosis in glioma, with STAT3 playing a role in this feedback loop [193]. Additionally, IGF2BP2 stabilizes transcripts like NUP214 and FLOT1, influencing glioma progression and TMZ resistance [120,124,194,195,196,197,198]. In glioma endothelial cells, IGF2BP3, in combination with METTL3, stabilizes CPEB2, maintaining the blood–tumor barrier and thereby influencing drug delivery [199]. It is also targeted by YTHDF2, influencing oncogene expression in GSCs [164]. Compared to other IGF2BPs, IGF2BP3 shows a higher correlation with stemness markers and immune infiltration in gliomas. Knockdown studies in glioma cells reveal that IGF2BP3 is integral to cell proliferation, invasion, and migration [200].

Another important m6A reader is eukaryotic initiation factor 3 (eIF3), which is crucial in the initiation of mRNA translation and exhibits significant implications in gliomas. eIF3 is unique in its ability to recognize m6A modifications on the 5′UTR of mRNA. This recognition allows for the initiation of mRNA translation independent of the conventional 5′ 7-methylguanosine (m7G) cap recognition by eukaryotic initiation factor 4E (eIF4E). eIF3 recruits the 43S preinitiation complex, which includes the 40S ribosomal subunit and eIF1A, eIF1, eIF2, and eIF3, initiating the translation of m6A-modified mRNA [201]. In U251 glioma cells, the knockdown of eIF3 subunit e led to an increase in mRNAs related to the p53 pathway, such as FAS and GADD45α, and a decrease in mRNAs related to survival and DNA replication/repair, such as UBE2V1, FGF11, CDC45, and JAK3. Furthermore, silencing of the eIF3 subunit e resulted in increased radiosensitivity in LN18 and U251 glioblastoma cell lines, suggesting a potential therapeutic avenue [202]. Targeting the eIF3 subunit c in U87 cells via siRNA led to suppressed cell proliferation, reduced colony formation, arrested cell cycle progression at the G0/G1 phase, promoted apoptosis, and prevented tumorsphere formation in xenograft models. Similar effects were observed in U251 cells, where knockdown of the eIF3 subunit c decreased proliferation and increased apoptosis [203,204]. Studies targeting the eIF3 subunit b in U87 cells and subunit d in both U87 and U251 cells showed that the knockdown of these subunits impeded glioma cell growth and proliferation, indicating their crucial role in glioma cell survival [205,206]. While not directly related to m6A methylation reading, the various subunits of eIF3 have potential prognostic value in gliomas. The expression levels of these subunits correlate with the severity and progression of the disease, suggesting their usefulness as biomarkers for glioma prognosis [207]. eIF3, particularly its various subunits, plays a significant role in the molecular mechanisms of glioma, influencing cell proliferation, survival, and response to treatments such as radiotherapy. The ability of eIF3 to initiate translation of m6A-modified mRNA independent of the m7G cap adds another layer of complexity to its role in glioma biology, presenting potential targets for therapeutic intervention. Targeting m6A reader proteins, like eIF3, to enhance radiotherapy sensitivity and modulate YTHDF, YTHDC, and HNRNP proteins to counteract chemotherapeutic resistance represents future research avenues to improve glioma treatment efficacy. Additionally, investigating the IGF2BP family’s influence on stemness and immune evasion could lead to novel interventions. The development of biomarkers to quantify these proteins’ expression will facilitate personalized treatment approaches.

### 4.4. Other RNA Modifications in Glioma Pathogenesis and Treatment Resistance

Knowledge of other RNA modifications in glioma pathogenesis is still limited. Some reports have demonstrated the influence of RNA m5C modifications on glioma’s biological characteristics, including proliferation, differentiation, migration, and malignancy [208]. Regulators of m5C, including NOP2, NSUN4, NSUN5, and NSUN7, have been linked to poor prognosis, while NSUN6 is associated with better outcomes [209]. In the glioblastoma cell line U87, NSUN2 has been shown to facilitate cell migration through the enhancement of mRNA cytosine methylation [210]. Compared to healthy tissue, glioma specimens show varied expression of m5C regulators, including DNMTs, NSUNs, TETs, YTHDF2, ALYREF, and YBX1 [211].

A-to-I editing is another significant RNA modification in glioma. It involves the re-editing of the glutamate receptor subunit B at the Q/R site, crucial for receptor functionality in the central nervous system [212]. Low-activity ADAR2, prevalent in gliomas, leads to reduced RNA editing of the GluA2 subunit at the Q/R site [213,214]. Similarly, ADAR3, a brain-specific adenosine deaminase, plays a comparable role in gliomas [215]. Decreased ADAR3 expression correlates with glioma progression, and A-to-I editing has been extensively linked to glioblastoma cell proliferation, migration, and invasion [216,217,218]. In terms of treatment resistance, experimental findings have revealed that increased expression of ADAR3 enhances the resistance to TMZ [219]. This expression alteration affects 641 genes, primarily regulated by NF-κB signaling pathways. Additionally, GSCs are central to A-to-I RNA editing in glioblastoma, with ADARs influencing GSC self-renewal and stem-like traits, potentially affecting the response to TMZ.

The role of other RNA modifications, such as m7G RNA methylation, in glioma, is still being unraveled. Research has indicated that out of 31 m7G methylation regulators studied, 17 exhibited higher expression levels in gliomas [220]. Moreover, an imbalance in m1A regulators is closely related to glioma onset and progression [220,221]. The inhibition of TRMT6, an m1A methyltransferase, impairs glioma cell proliferation, migration, and invasion (Figure 2).

## 5. Conclusions

In summary, the rapidly advancing field of epitranscriptomics, particularly RNA modifications, has gained significant attention as a novel focus in understanding and potentially targeting neurological diseases. This review has highlighted the essential role of RNA modification in various diseases, emphasizing its impact on mRNA stability, translation, and the control of protein levels in disease-associated pathways. Specifically, m6A modification in glial cells is crucial for the onset and progression of neurological diseases like Alzheimer’s disease and glioma. However, despite the current focus on DNA methylation, histone modifications, and chromatin rearrangement in neurological diseases, the critical biological functions of RNA modification have been relatively underexplored and warrant further investigation. Additionally, while m6A modification is increasingly studied in various fields, a significant number of enzymes responsible for these modifications—known as writers, erasers, and readers—remain unidentified. This holds significant pertinence when considering neurodegenerative diseases like Parkinson’s disease (PD), where m6A-modification genes have been identified, but their association with the disease needs more comprehensive study. Therefore, future efforts should delve deeper into the role of RNA modifications in the nervous system, which could reveal new molecular targets for pharmacological and clinical therapy development for uncurable neurological diseases.

## Figures and Tables

**Figure 1 cells-13-00457-f001:**
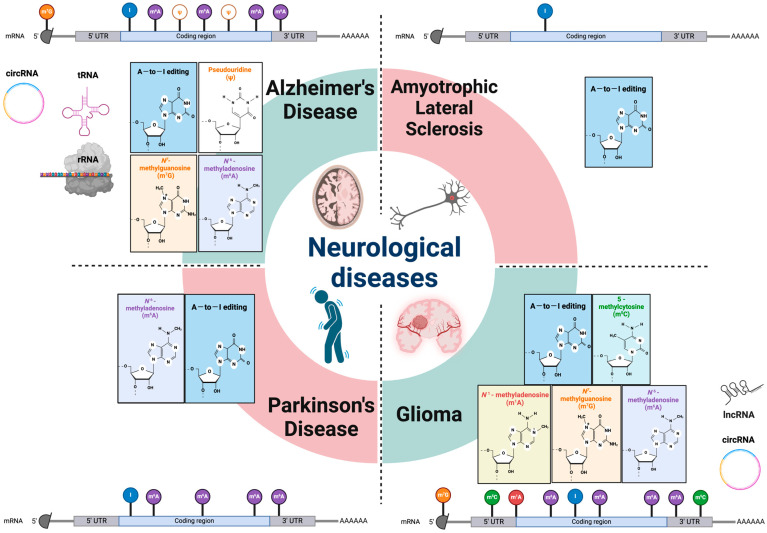
RNA modifications in various neurological diseases. Epitranscriptomic RNA modifications in both mRNA and different non-coding RNAs are involved in multiple types of neurologic disorders. Genetic analysis and animal model studies have revealed that various RNA modifications and their machineries have critical roles in the etiology of neurodegenerative diseases and gliomas. mRNA: messenger RNA; rRNA: ribosomal RNA; lncRNA: long non-coding RNA; tRNA: transfer RNA; circRNA: circular RNA; UTR: untranslated region; A-to-I editing: adenosine to inosine RNA editing.

**Figure 2 cells-13-00457-f002:**
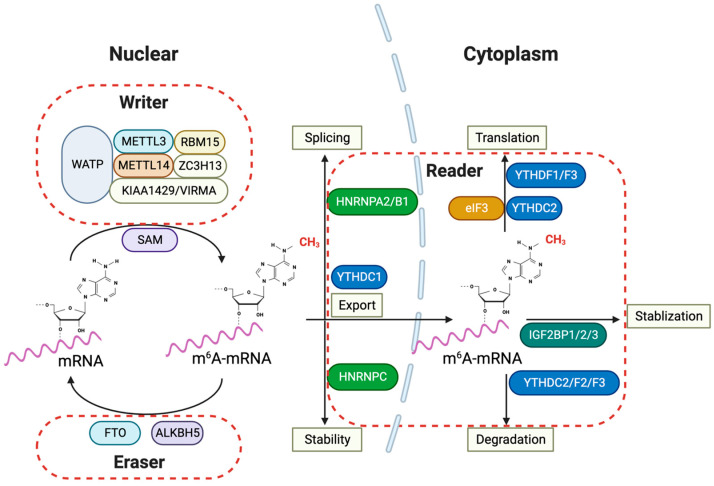
m6A RNA modification and its regulatory proteins in glioma pathogenesis. The m6A RNA modification pathways implied for glioma pathogenesis involve various nuclear and cytoplasmic processes in the cells. The ‘writers’ of the m6A mark, which include METTL3, METTL14, WTAP, RBM15, ZC3H13, and KIAA1429/VIRMA, collectively form the methyltransferase complex in the nucleus, utilizing SAM as a methyl donor to catalyze mRNA methylation. Conversely, ‘erasers’ such as FTO and ALKBH5 demethylate m6A-mRNA, reversing the modifications introduced by the writers. Within the cytoplasm, ‘readers’ such as YTHDFs, YTHDC2, HNRNPC, and the IGF2BP family recognize and bind to m6A-mRNA, affecting its stability, nuclear export, splicing, and translation. The involvement of IGF2BPs indicates their role in stabilizing m6A-mRNA, which is crucial for regulating gene expression that is vital for glioma progression. Furthermore, proteins like HNRNPs and eIF3 are implicated in splicing and translation, underscoring the complexity and importance of the m6A modification system in glioma pathogenesis. METTL3: Methyltransferase Like 3; METTL14: Methyltransferase Like 14; WTAP: Wilms’ Tumor 1-Associating Protein; RBM15: RNA Binding Motif Protein 15; ZC3H13: Zinc Finger CCCH-Type Containing 13; KIAA1429/VIRMA: Vir Like m6A Methyltransferase Associated; SAM: S-adenosylmethionine; FTO: Fat Mass and Obesity-Associated Protein; ALKBH5: AlkB Homolog 5; YTHDF: YTH Domain Family; YTHDC: YTH Domain Containing; HNRNP: Heterogeneous Nuclear Ribonucleoprotein; IGF2BP: Insulin-like Growth Factor 2 mRNA-Binding Proteins; eIF3: Eukaryotic Translation Initiation Factor 3.

**Table 1 cells-13-00457-t001:** The role of RNA modifications on diverse RNA metabolic activities and their association with neurological diseases.

	Neurodegen Erative Diseases	RNA Modifications	The Role in RNA Metabolism	Reader/Writer/Eraser	Mechanism	Detection
Neurodegenerative diseases	PD	m6A	Trasnlation	Eraser: FTO	In neurons affected by disease, FTO is produced and accumulates in the axons, leading to enhanced demethylation of m6A and increased expression of NMDAR1, which is followed by neuronal cell death.	
	AD		Translation	Writer: METTL3 Eraser: FTO	mRNA methylation regulates the expression of transcripts related to AD	
	AD	Pseudouridine	mRNA stability, translation	Writer: PUS1	Unknown. Only the reduction of pseudouridine in AD cortex has been reported.	
	AD, PD, ALS	RNA editing	Transport/Translation	Writer: ADAR2	The pre-mRNA of the AMPA receptor is modified by ADAR2 to control its activity. A decrease in ADAR2 levels leads to Iess editing, which in turn is associated with functional impairments of the AMPA receptor under pathological conditions.	
Glioma		m6A	Transport, localization, translation, and splicing	Writers: METTL3, METTL14, WTAP, KIAA1429/VIRMA, RBM15, and ZC3H13 Erasers: FTC and ALKBH5 Readers: YTHDF1/F3, YTHDC1, YTHDC2, HNRNPA2/B1, HNRNPC, IGF2BP1/2/3,	Writers: Affect the stability, transport, localization, translation, and splicing of RNA, thereby contributing to gene expression regulation. Erasers: Demethylate modified adenosine residues, altering RNA molecules’ fate and regulating their expression patterns and functions. Readers: Modify the stability, localization, and translation efficiency of RNA, thereby regulating its function and intracellular dynamics and fine-tuning gene expression.	LC-MS/MS, MeRIP-Seq, qRT-PCR, IP-MS,
		m5C	Stability and translation	Writers: DNMT2 and NSUN Readers: ALYREF and YBX1	Writers: Affect the stability, transport, localization, translation, and splicing of RNA, thereby contributing to gene expression regulation. Readers: Modify the stability, localization, and translation eficiency of RNA, thereby regulating its function and intracellular dynamics and fine-tuning gene expression.	LC-MS/MS, RNA-BisSeq
		A-to-I editing	Coding potential and splicing	ADAR1 and ADAR2		LC-MS/MS, DAR enzymes

m6A: N6-Methyladenosine; m5C: 5-Methylcytosine; A-to-I editing: adenosine to inosine RNA editing; AD: Alzheimer’s Disease; PD: Parkinson’s Disease; ALS: Amyotrophic Lateral Sclerosis; METTL3: Methyltransferase Like 3; METTL14: Methyltransferase Like 14; WTAP: Wilms’ Tumor 1-Associating Protein; RBM15: RNA Binding Motif Protein 15; ZC3H13: Zinc Finger CCCH-Type Containing 13; KIAA1429/VIRMA: Vir Like m6A Methyltransferase Associated; SAM: S-adenosylmethionine; FTO: Fat Mass and Obesity-Associated Protein; ALKBH5: AlkB Homolog 5; YTHDF: YTH Domain Family; YTHDC: YTH Domain Containing; HNRNP: Heterogeneous Nuclear Ribonucleoprotein; IGF2BP: Insulin-like Growth Factor 2 mRNA-Binding Proteins; eIF3: Eukaryotic Translation Initiation Factor 3; ADAR: adenosine deaminase acting on RNA; PUS: pseudouridine synthase; DNMT: DNA methyltransferase; NSUN: NOP2/SUN RNA methyltransferase; YBX1: Y-box binding protein1; LC-MS/MS: Liquid chromatography–tandem mass spectrometry; MeRIP-Seq: Methylated RNA immunoprecipitation sequencing; RNA-BisSeq: RNA bisulfite sequencing; qRT-PCR: quantitative Real-Time PCR; IP-MS: immunoprecipitation coupled with mass spectrometry.

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
