# Peer review of "Emerging Roles and Mechanisms of RNA Modifications in Neurodegenerative Diseases and Glioma"

_cells, 2024, doi:10.3390/cells13050457_

Round 1

Reviewer 1 Report

Comments and Suggestions for Authors

In this review, the authors have discussed the epitranscriptomics and its role in neurological diseases. The authors discussed about various RNA modification such as m6A, m1A, m5C, pseudouridine, and RNA editing (A to I). The manuscript is well written and properly organized. The references are recent. The authors have critically discussed the roles of these modification in various diseases. I have few minor comments.

1. The authors should also describe about the RNA modification of circular RNA and discussed its role in neurological disease.

2. The authors should add a short paragraph on various method for detecting the various RNA modification.

3. The authors should add a figures showing all the RNA modifications. 

Reviewer 2 Report

Comments and Suggestions for Authors

This is an interesting and compelling review that highlights the importance of RNA editing in brain malfunctioning and refers to many levels of such editing processes, which are clear virtues of this manuscript. The lack of specific explanations on the methods involved in analyzing the different modifications involved; a text box including specific advice on such methods may add to the visibility of this review. Also, the part dedicated to brain tumors, especially glioblastoma is so large that it deserves mentioning in the title for attracting the attention of specific readers.
